# Hydrochemical Characteristics and Ion Sources of Precipitation in the Upper Reaches of the Shiyang River, China

Zhiyuan Zhang [1] , Wenxiong Jia [1,*], Guofeng Zhu [1,2,3], Xinggang Ma [2], Xiuting Xu [1], Ruifeng Yuan [1], Yang Shi [1], Le Yang [1] and Hui Xiong [1]

1   College of Geography and Environmental Science, Northwest Normal University, Lanzhou 730070, China; zhiyuanZhang1325@126.com (Z.Z.); zhugf@nwnu.edu.cn (G.Z.); xxtais@163.com (X.X.); 18709480326@163.com (R.Y.); syang0322@163.com (Y.S.); yang_le13@163.com (L.Y.); xiong673677431@163.com (H.X.)

2   State Key Laboratory of Cryosphere Science, Northwest institute of Eco-Environment and Resources, Chinese Academy of Sciences, Lanzhou 730070, China; xgmaxg@126.com

3   Gansu Engineering Research Center of Land Utilization and Comprehension Consolidation, Lanzhou 730070, China

*   Correspondence: jwxiong@nwnu.edu.cn

**Abstract:** The Shiyang River Basin is located at the edge of the monsoon wind system of South and Southeast Asia. The hydrochemical characteristics of precipitation are influenced by both monsoon and arid regions. The regression analysis method, comparative analysis, neutralization factor (NF), enrichment factor (EF) and HYSPLIT4 were used to analyze the precipitation samples collected from the upper reaches of the Shiyang River from October 2016 to October 2017. In order to study the hydrochemical characteristics and ion sources of precipitation in this basin. The results, as discussed in this paper, show that the precipitation in the upper reaches of the Shiyang River is mildly alkaline all year round while the neutralization ability of $Ca^{2+}$ and $NH_4^+$ in precipitation is strong. The ion concentration was higher in the dry season than that in the wet season, but the concentration of $NH_4^+$ was higher in summer. Furthermore, as the altitude increased, the electrical conductivity (EC) of the precipitation decreased gradually. Influenced by precipitation and rainy days, the wet deposition of nitrogen (N) and sulfur (S) was higher in the wet season than that during the dry season, and the wet deposition gradually increased with the elevation. In precipitation, the earth's crust is a major source of $Ca^{2+}$, $K^+$ and $Mg^{2+}$, the ocean is a major source of $Cl^-$, the ocean and the earth's crust are the sources of $Na^+$, human activities are the main sources of $SO_4^{2-}$, $NO_3^-$ and $NH_4^+$, the amount of $F^-$ is very small, its sources are natural and human activities. Water vapor in precipitation mainly comes from westerly air mass circulation and monsoon circulation while the particles come mainly from the earth's crust.

**Keywords:** Shiyang River; precipitation; ion; wet deposition; enrichment factor method

## 1. Introduction

Atmospheric precipitation is an important component of the water cycle of the earth. It can bring aerosol particles such as dust, smoke particles, water droplets, salt particles, etc. in the atmosphere to the ground by the way of wet deposition, effectively reducing the particles in the atmosphere, and thereby purifying the atmosphere [1–3]. By collecting and analyzing the ion composition in precipitation, the changes in composition of atmospheric aerosol particles can be studied, so as to monitor atmospheric pollution effectively [4–10]. In recent decades, China's GDP has been growing

continuously, and the rapid development of industrialization has also aggravated and increased the consumption of energy. China has become the third largest acid rain area in the world after Europe and North America [11]. Domestic research on precipitation focuses mainly on the situation of acid rain in the central and eastern regions of China [12–20]. These regions have a high level of urbanization and rapid industrial development, and have relatively serious air pollution. The precipitation is mainly acidic and the leading pollutants are $NO_3^-$ and $SO_4^{2-}$. However, the western region is populated sparsely and is relatively backward with respect to the level of industrial development, and as such is less affected by acid rain. The annual precipitation in the northwest arid region is less than 200 mm. Most of the soil sources in this area are alkaline deserts and sandy land, and the vegetation is sparse. Under the influence of westerly air mass circulation and cold anticyclone, dusty weather is common, so there are relatively more alkaline ions in the atmosphere.

The Shiyang River Basin, located between the Qinghai–Tibet Plateau and the Tengger Desert, is one of the three inland river basins in the Hexi region. It is the main source of industrial and agricultural production, urban development and oasis irrigation in Wuwei, Minqin and other human settlements in the middle and lower reaches of this basin. Precipitation as an important way of replenishing a river, the pH, electrical conductivity (EC) and chemical ion composition of precipitation have a significant impact on the ecology and water resources of this region [21–28].

Although there have been relevant studies on precipitation in this region, there are few observations and studies on specifically the upstream regions. This paper is based on the research done from seven sample points in the upper reaches of the Shiyang River region. Samples for the study were collected from October 2016 until October 2017. The chemical composition and temporal and spatial variation characteristics of precipitation were compared and analyzed. Using the enrichment factor method and the HYSPLIT4 to analyze ion and water vapor sources of precipitation, the purpose of the study was to understand the hydrochemical characteristics, ion composition and water vapor source of precipitation in the study area. In this paper, on the basis of existing studies, the neutralization factor method was added to study the influence of cation on precipitation pH, and wet deposition was calculated to explore the environmental pollution in this area, thus providing a theoretical basis for the treatment of the ecological environment.

## 2. Study Area

The Shiyang River Basin (101°22′–104°14′ E, 37°7′–39°27′ N) is located in the northern section of the Qilian Mountain, the southern of the Tengger Desert, the eastern of the Hexi Corridor and the western section of the Wushaoling Mountains, covering an area of about $4.16 \times 10^4$ km$^2$ (Figure 1). The terrain is high in the south and low in the north and comprises the Qilian Mountains, the corridor plains and the low hills and the desert areas from the south to the north. The water system of the Shiyang River consists of the Dajing River, the Gulang River, the Huangyang, the Zamu, the Jinta, the Xiying, the Dongda, the Xida Rivers and other smaller rivers from east to west. The average annual runoff is $1.56 \times 10^9$ m$^3$, and the primary source of river water supply is from atmospheric precipitation and melting snow from the Qilian Mountains. The Shiyang River Basin has a typical temperate continental arid climate, and has vertical zonality due to the influence of the topography of the region.

The south of the basin is composed of Cambrian, Ordovician and Silurian metasandstone, slate rocks, clastic rocks, carbonate rocks, intermediate-acidic volcanic rocks, intermediate-basic volcanic rock and magmatic rocks. The middle of the basin is mainly composed of Sinian, pre-Sinian, Cambrian gneiss, gneiss rocks, phyllite rocks, slate rocks, metamorphic sandstone, carbonate rocks, intermediate-basic volcanic rocks and granite. The northern part of the basin is mainly composed of Paleozoic magmatic rocks. The rocks in the study area are mainly carbonate rocks and silicate rocks, whose composition is mainly plagioclase, calcite and dolomite, mainly including $Ca^{2+}$, $Mg^{2+}$ and $Na^+$ [29].

It is divided into three regions from south to north: (1) the alpine semi-arid and humid region of the Qilian Mountains that has an altitude of 2000–5000 m, an annual average temperature of less than 0 °C, an annual average precipitation of 300–600 mm and a potential annual evaporation rate

of 700–1200 mm; (2) the central corridor that has an altitude of 1500–2000 m, an annual average temperature of less than 7.8 °C, an annual precipitation of 150–300 mm and a potential annual evaporation rate of 1300–2000 mm; (3) the warm and arid region in the north that has an altitude of 1300–1500 m, an annual average temperature of less than 8 °C, an annual precipitation below 150 mm and a potential annual evaporation rate of 2000–2600 mm. Vegetation and soil also have vertical zonality. From south to north, the vegetation types are meadow, shrub meadow, forest, grassland and desert, correspondingly the soil types are meadow soil, shrub meadow soil, gray cinnamon soil, chestnut soil and gray calcium soil.

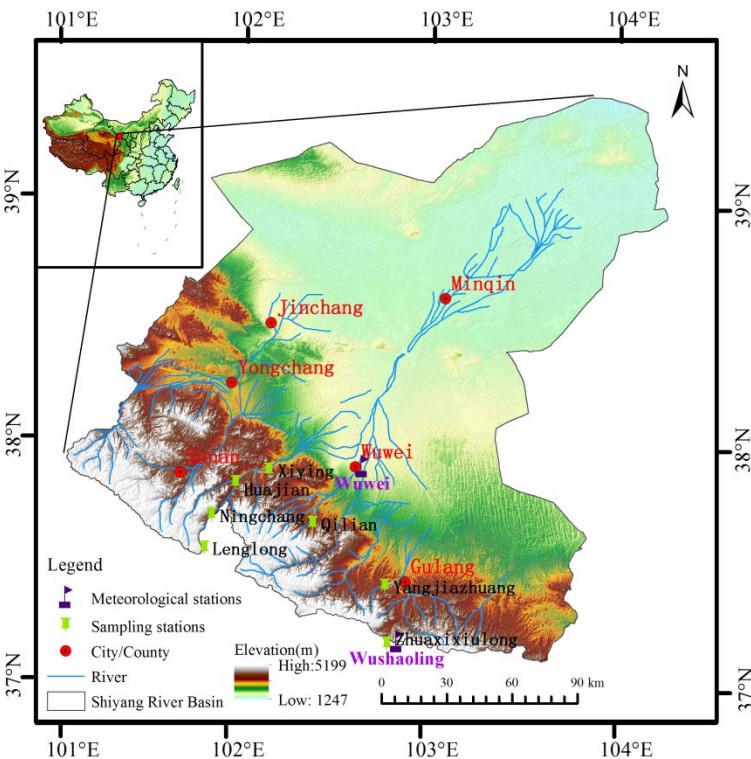

**Figure 1.** Geographical location and sampling point distribution of the study area.

## 3. Data and Method

### 3.1. Experimental Design and Precipitation Sampling

From October 2016 until October 2017, 355 precipitation samples (Table 1) were collected continuously from 7 sampling points (Qilian only recorded precipitation from June to August). Using the standard rain gauge, each of the collected precipitation was poured into a polyethylene sample bottle and capped after the end of a precipitation event. Samples were shipped to the Ecological and Hydrological Process Laboratory of Northwest Normal University and stored in a freezing laboratory (about −15 °C). Forty-eight hours before being tested, the samples were taken out and placed at room temperature (about 21 °C) for natural melting. Thereafter the pH, the EC and the main ion concentrations were determined at the National Key Laboratory of Cryosphere Science, Cold and Arid Regions Environmental and Engineering Research Institute, Chinese Academy of Sciences (CAREERI, CAS). EC and pH were determined by means of the Seven Excellence$^{TM}$ (Shanghai Lianxiang Environmental Protection Technology Co., Ltd., Shanghai, China). The measurement range of EC was between 0.001 and 2000 μs/cm, with an accuracy of ±0.5%, and that of pH is from 0.000 to 14.000 pH, with an accuracy of ±0.05%. The concentrations of $Na^+$, $K^+$, $Mg^{2+}$, $Ca^{2+}$ and $NH_4^+$ were determined by means of the DIONEX DX320 ion chromatograph (DIONEX Co., Ltd., Sunnyvale, CA, USA), and those of $Cl^-$, $F^-$, $NO_3^-$ and $SO_4^{2-}$ by means of the DIONEX ICS1500 ion chromatograph

(DIONEX Co., Ltd., Sunnyvale, CA, USA). The accuracy of these can reach ng/g while the test data error does not exceed 5%.

**Table 1.** Sampling location and sample quantity.

| Sampling Point | Longitude (E) | Latitude (N) | Elevation (m) | Precipitation (mm) | Number of Samples | Spring | Summer | Autumn | Winter |
|---|---|---|---|---|---|---|---|---|---|
| Lenglong | 101.86° | 37.56° | 3648 | 1025.12 | 94 | 31 | 32 | 26 | 5 |
| Ningchang | 101.89° | 37.70° | 2721 | 469.44 | 56 | 17 | 15 | 22 | 2 |
| Huajian | 102.01° | 37.83° | 2323 | 282.05 | 51 | 15 | 23 | 13 | |
| Xiying | 102.18° | 37.89° | 2097 | 197.67 | 47 | 7 | 23 | 15 | 2 |
| Yangjiazhuang | 102.80° | 37.42° | 2356 | 613.54 | 46 | 15 | 16 | 11 | 4 |
| Zhuaxixiulong | 102.82° | 37.19° | 2858 | 290.63 | 51 | 18 | 17 | 10 | 6 |
| Qilian | 102.42° | 37.68° | 2379 | 60.5 | 10 | | 10 | | |

In order to take into account, the influence of the amount of rainfall on the concentration of ions, the volume-weighted mean (VWM) and standard deviation of the volumetric mean (VWSD) of precipitation in different locations and seasons was calculated to modify the data [5,9,30–32]. The calculation method and the results (Table 2) are as follows:

$$\overline{pH} = \frac{\sum\limits_{i=1}^{n} pH_i \times P_i}{\sum\limits_{i=1}^{n} P_i} \tag{1}$$

$$\overline{EC} = \frac{\sum\limits_{i=1}^{n} EC_i \times P_i}{\sum\limits_{i=1}^{n} P_i} \tag{2}$$

$$\overline{C} = \frac{\sum\limits_{i=1}^{n} C_i \times P_i}{\sum\limits_{i=1}^{n} P_i} \tag{3}$$

where $\overline{pH}$, $\overline{EC}$ and $\overline{C}$ is the VWM of the precipitation of pH, EC and ion concentration, respectively. $P_i$ represents the precipitation (mm) of the $i$ sample. $pH_i$, $EC_i$ (μs/cm) and $C_i$ (μeq/L), respectively represent the pH, EC and ion concentration of the $i$ sample. According to the climate characteristics of the study area, the wet season was defined as from April to September and the dry season from October to March of the following year.

**Table 2.** Chemical characteristics and main ion concentration of the precipitation water.

| Sampling Point | | EC | pH | Na$^+$ | NH$_4^+$ | K$^+$ | Mg$^{2+}$ | Ca$^{2+}$ | F$^-$ | Cl$^-$ | SO$_4^{2-}$ | NO$_3^-$ |
|---|---|---|---|---|---|---|---|---|---|---|---|---|
| Lenglong | Min | 7.15 | 6.56 | 4.38 | 11.85 | 1.48 | 0.48 | 8.54 | 0.54 | 0.82 | 0.49 | 0.13 |
| | Max | 339.00 | 8.60 | 756.10 | 273.27 | 90.14 | 1799.25 | 1752.21 | 12.93 | 255.72 | 856.06 | 32.42 |
| | VWM | 42.17 | 7.41 | 56.88 | 55.67 | 15.52 | 25.36 | 160.73 | 0.57 | 20.07 | 61.94 | 5.03 |
| | VWSD | 1.31 | 0.72 | 0.21 | 0.79 | 2.28 | 0.02 | 0.53 | 1.04 | 0.06 | 0.43 | 0.06 |
| Ningchang | Min | 9.38 | 6.89 | 2.31 | 8.16 | 0.92 | 2.21 | 15.86 | 0.53 | 1.65 | 1.23 | 0.62 |
| | Max | 484.00 | 8.68 | 1042.02 | 832.78 | 837.42 | 792.75 | 2023.50 | 6.43 | 404.83 | 643.05 | 174.83 |
| | VWM | 32.31 | 7.63 | 39.26 | 71.47 | 20.27 | 26.61 | 126.46 | 0.13 | 17.83 | 21.52 | 8.82 |
| | VWSD | 0.55 | 0.10 | 1.37 | 1.36 | 0.83 | 0.87 | 2.83 | 0.01 | 0.54 | 0.69 | 0.23 |
| Huajian | Min | 16.50 | 6.54 | 4.78 | 45.55 | 0.51 | 2.50 | 31.50 | 0.42 | 3.38 | 8.33 | 1.61 |
| | Max | 305.00 | 8.21 | 683.71 | 1672.78 | 340.80 | 624.74 | 1332.46 | 7.37 | 272.25 | 598.80 | 95.47 |
| | VWM | 49.99 | 7.21 | 45.21 | 597.06 | 20.11 | 39.48 | 150.62 | 0.75 | 22.25 | 45.09 | 11.34 |
| | VWSD | 0.96 | 0.13 | 0.93 | 21.20 | 0.69 | 0.94 | 2.64 | 0.03 | 0.49 | 0.90 | 0.23 |
| Xiying | Min | 10.04 | 6.93 | 1.37 | 0.00 | 1.12 | 1.78 | 14.17 | 0.72 | 1.49 | 3.12 | 0.68 |
| | Max | 134.30 | 8.03 | 369.83 | 218.87 | 139.68 | 463.88 | 631.54 | 3.98 | 167.55 | 220.26 | 100.44 |
| | VWM | 40.71 | 7.32 | 42.80 | 73.47 | 15.20 | 37.55 | 145.46 | 0.27 | 20.02 | 37.32 | 14.86 |
| | VWSD | 0.54 | 0.07 | 1.01 | 0.97 | 0.35 | 1.07 | 2.07 | 0.01 | 0.43 | 0.63 | 0.27 |
| Yangjiazhuang | Min | 6.59 | 6.86 | 0.43 | 30.25 | 0.26 | 0.83 | 3.50 | 0.38 | 0.56 | 1.88 | 0.81 |
| | Max | 152.10 | 8.44 | 833.76 | 1598.89 | 360.95 | 310.92 | 1106.73 | 17.57 | 370.45 | 262.69 | 63.57 |
| | VWM | 25.29 | 6.57 | 29.22 | 455.35 | 12.64 | 13.60 | 89.97 | 0.48 | 15.75 | 27.43 | 10.53 |
| | VWSD | 0.57 | 0.14 | 0.81 | 19.24 | 0.62 | 0.43 | 3.11 | 0.04 | 0.50 | 0.80 | 0.35 |
| Zhuaxixiulong | Min | 15.27 | 6.92 | 9.50 | 42.44 | 3.65 | 2.46 | 17.82 | 0.27 | 4.86 | 5.48 | 0.68 |
| | Max | 476.00 | 8.96 | 1351.15 | 344.70 | 2101.38 | 851.66 | 1054.81 | 5.83 | 748.19 | 642.80 | 69.19 |
| | VWM | 57.06 | 7.69 | 69.34 | 97.39 | 40.46 | 41.60 | 167.08 | 0.43 | 29.03 | 45.25 | 11.57 |
| | VWSD | 0.97 | 0.12 | 1.89 | 1.61 | 1.37 | 0.79 | 3.29 | 0.01 | 0.77 | 0.94 | 0.31 |

Notes: the max and min values are the experimental values of a single sample. VWM is a volume-weighted mean by region.

### 3.2. Research Methods

#### 3.2.1. Law of Conservation of Electric Charge

According to the law of charge conservation, the total charge of all negative anions in each sample was the same as that of cations. In order to verify the data quality of the samples, the law of conservation of the electric charge of 355 samples was applied. By converting the anion and cation units from mg/L to μeq/L, and then performing the linear-regression analysis, we arrived at the regression equation $y = 3.46x + 23.83$, $R^2 = 0.96616$, $p < 0.05$ (Figure 2), cations and anions were significantly related. $CO_3^{2-}$ and $HCO_3^{-}$ were found in the most natural water. When $6.4 < pH < 10.3$, c $[H_2CO_3] <$ c $[HCO_3^{-}] >$ c $[CO_3^{2-}]$, and when $pH < 8.3$, the content of $CO_3^{2-}$ can be ignored [33]. The reason why the regression equation deviated from the standard equation was because important anions such as $CO_3^{2-}$ and $HCO_3^{-}$ were missing, therefore the charge of the cations was significantly higher than that of the anions.

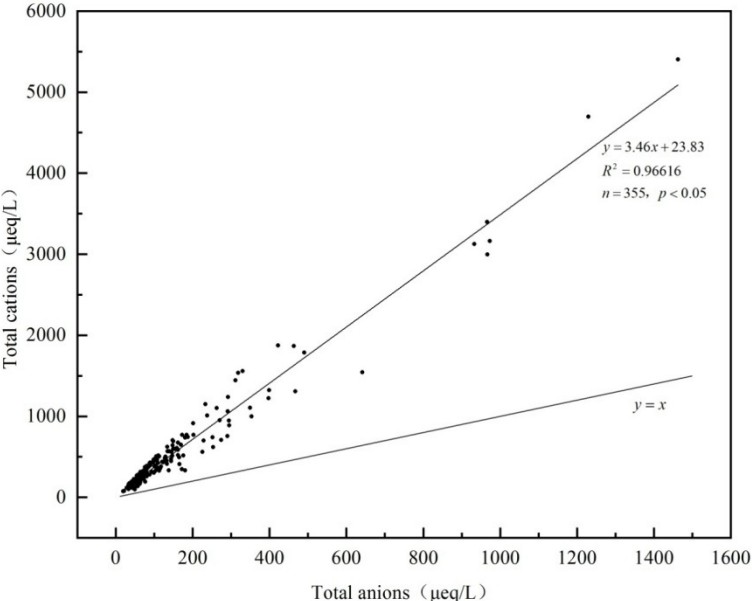

**Figure 2.** Scatter plot of total anions versus total cations.

#### 3.2.2. Neutralization Factor (NF) Method

Sand and dusty weather and human activities are all common in the Shiyang River Basin, therefore many alkaline ions are present in the air, such as $Ca^{2+}$ and $NH_4^{+}$, the presence of these ions has a certain effect on the pH of the precipitation. In order to evaluate the neutralization ability of alkaline ions, the neutralization factor $(NF)_X$ was calculated by means of the following formula [34]:

$$(NF)_X = \frac{[X]}{[NO_3^-] + [SO_4^{2-}]} \tag{4}$$

where, $X$ represents the neutralization ion concentration (μeq /L).

#### 3.2.3. Enrichment Factor (EF) Method

To determine and evaluate the source of the element in the particulate matter, the enrichment factor method was used to express the elemental abundance in atmospheric particulate matter by comparing the ratio of ions in the reference object to the reference material. This method was first proposed by Keene [35].

On the basis of this information, Xiao summarized the reasonable selection method of the sea salt source indicator [36]. If the equivalent ratio of $Cl^-/Na^+$ and $Mg^{2+}/Na^+$ is equal or greater to the corresponding value of seawater ($Cl^-/Na^+ = 1.165$, $Mg^{2+}/Na^+ = 0.227$), cations of $Na^+$ are used as the indicator of sea salt source. If the equivalent ratio of $Na^+/Cl^-$ and $Mg^{2+}/Cl^-$ is equal or greater than the corresponding value of seawater ($Na^+/Cl^- = 0.859$, $Mg^{2+}/Cl^- = 0.195$), then anions of $Cl^-$ are used as the indicator of sea salt source. If the equivalent ratio of $Na^+/Mg^{2+}$ and $Cl^-/Mg^{2+}$ is equal or greater than the corresponding value of seawater ($Na^+/Mg^{2+} = 4.403$, $Cl^-/Mg^{2+} = 5.126$), then cations of $Mg^{2+}$ are selected as the sea salt source indicator.

According to the selection method of this indicator, the anion of $Cl^-$ was selected as the sea salt source indicator in the study area under discussion here, because the equivalent ratio of $Na^+/Cl^-$ was 2.476 (range 1.856–2.834) and that of $Mg^{2+}/Cl^-$ was 1.575 (range 0.863–1.876). Since it is abundantly present in the earth's crust and its composition is not changed easily, the element of Ca was used as a reference element for terrestrial sources. According to the chart of Gibbs, there are two main sources of rainfall and rock differentiation for the solute of land water. If the contributions of volcanoes and other natural sources are ignored, the main sources of the precipitation ion composition in the atmosphere include seawater sputtering, rock and soil weathering and human activities. In order to explore the contribution rate of different factors to ion sources, the relative contributions of marine input (*SSF*), rock and soil weathering (*CF*) and human activity input (*ASF*) were calculated. The calculation formula was used as follows:

$$(EF)_{sea} = \frac{\left[\frac{X}{Cl^-}\right]_{rain}}{\left[\frac{X}{Cl^-}\right]_{sea}} \tag{5}$$

$$(EF)_{crust} = \frac{\left[\frac{X}{Ca^{2+}}\right]_{rain}}{\left[\frac{X}{Ca^{2+}}\right]_{crust}} \tag{6}$$

$$SSF(\%) = \frac{\left[\frac{X}{Cl^-}\right]_{sea}}{\left[\frac{X}{Cl^-}\right]_{rain}} \times 100\% \tag{7}$$

$$CF(\%) = \frac{\left[\frac{X}{Ca^{2+}}\right]_{crust}}{\left[\frac{X}{Ca^{2+}}\right]_{rain}} \times 100\% \tag{8}$$

$$ASF(\%) = 100\% - SSF(\%) - CF(\%) \tag{9}$$

where *X* is the concentration of precipitation ions and the ratio of $[X/Cl^-]_{sea}$ and $[X/Ca^{2+}]_{crust}$ refers to the enrichment factor (EF) of each ion in precipitation [26,37,38]. If the value of EF is less than 1, it indicates that the ion is diluted. In contrast, if the value of EF is more than 1, it indicates that the ion is enriched, and therefore also indicates that the larger the enrichment factor, the higher the degree of enrichment.

### 3.2.4. Estimation of Wet Deposition of Ionic Species

Precipitation is an important source of accumulation of surface elements, the wet deposition of elements can indicate pollution and affect the natural environment and biosphere [12,31,39]. The annual wet deposition can be calculated as:

$$F_W = \sum_{i=1}^{n} C_i \times P_i \tag{10}$$

where $P_i$ represents the precipitation (mm) of the *i* and *Ci* (mg/L) represents the ion concentration of the *i* precipitation.

3.2.5. Backward Trajectory Method

In order to track the water vapor source in the study area, the HYSPLIT4 (hybrid single particle Lagrangian integrated trajectory) was used [40,41], that is the Lagrangian mixed single-particle orbit model (https://www.arl.noaa.gov/). This model is a professional model developed by the American Atmospheric Laboratory for calculating and analyzing the transport and diffusion trajectories of atmospheric pollutants. It has been used widely in the study of transmission and diffusion of various pollutants in various regions. In this study currently under discussion, the backward trajectory method was used to track the water vapor trajectory within 240 h, where the water vapor tracking height was 1500 m, to determine the source of water vapor.

## 4. Results

### 4.1. Precipitation pH and EC

pH is an indicator of acidity and alkalinity. Since the compound $CO_2$ in the atmosphere is readily soluble in water to form the compound $H_2CO_3$, which is ionized easily in water at normal temperature and pressure ($H_2CO_3 = HCO_3^- + H^+$), precipitation with a pH value of less than 5.6 is usually called acid rain. The range of pH in precipitation varied from 6.54 to 8.96 in the upper reaches of the Shiyang River (Figure 3). It is thus mildly alkaline all year around and is not affected by acid rain. The VWM of pH in precipitation were7.47, 7.24, 7.27 and 6.41 in spring, summer, autumn and winter respectively, which indicates that spring > autumn > summer > winter. The lowest value of pH appeared in October and the highest value of pH appeared in May. Precipitation events and precipitation are relatively low in winter, so the pH of precipitation was at its lowest then. Spring is a season of frequent dust weather. When dust weather occurs, it carries more alkaline ions, so the highest pH value appeared in spring. Frequent precipitation in summer and autumn dilutes the concentration of alkaline elements, so the pH value was low. Ammonia fertilizer in summer farmland easily decomposes when heated and since $NH_3$ is soluble in water and is alkaline, the pH value during summer was slightly higher during autumn. As most crops are harvested in autumn, the lowest pH value therefore appeared in autumn.

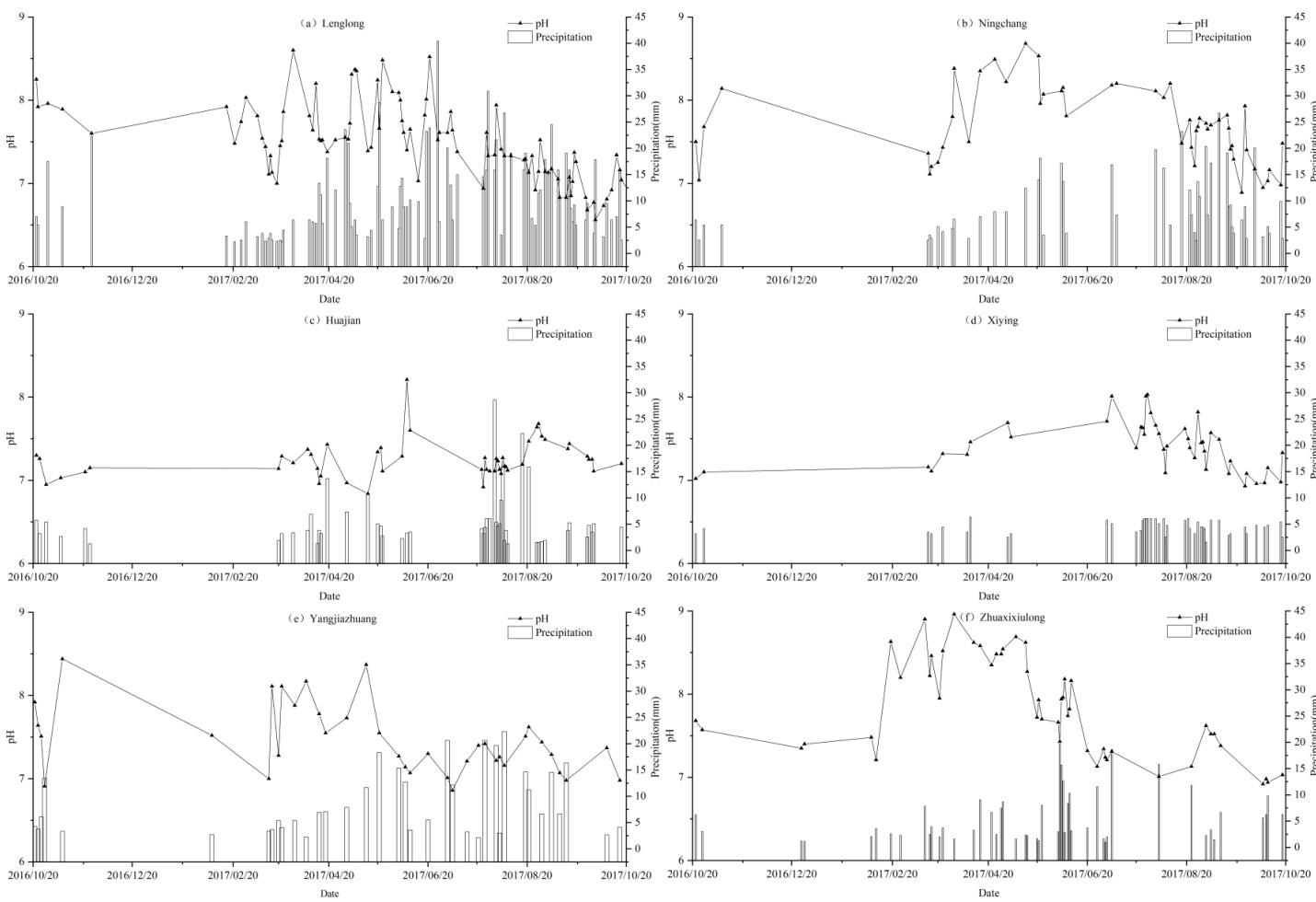

**Figure 3.** Seasonal variation in precipitation and pH.

The EC values of precipitation in the upper reaches of the Shiyang River were varied (Figure 4). When wet deposition occurred, the value of EC in precipitation was higher for a large amount of particulate matter in the atmosphere. The VWM of EC in precipitation during the dry season was 56.35 μs/cm, and during the wet season was 34.66 μs/cm, which indicates the value of EC was higher during the dry season than that during the wet season. This is consistent with the findings in the Tianshan Glacier Area of China [42–44]. It is known that the value of EC in precipitation is affected by the amount of precipitation, where more precipitation causes the value of EC to be low, and the reverse is also true. In the upper reaches of the Shiyang River, the precipitation in the dry season accounted for 16.33% of the annual precipitation, but it in the wet season it accounted for 83.67% of the annual precipitation. So, less precipitation in the dry season resulted in higher conductivity, but more precipitation in the wet season diluted the ion concentration and caused lower conductivity.

Sincethe Shiyang River Basin is close to the Tengger Desert, the value of EC in precipitation was higher in winter and spring due to the impact of the dusty weather (Figure 5, Table 3). According to the air quality index (AQI) of Wuwei City, the total number of pollution days (AQI ≥ 150) from October in 2016 to October in 2017 totaled 18 (78% of them happened the period from November to May), and the pollutants were mainly PM2.5 and PM10 (https://www.aqistudy.cn/historydata/). As shown in Table 3, the air quality of Wuwei City from November 5 to 6 in 2016 was slightly polluted (PM2.5 was 41 μg/m$^3$ and PM10 was 145 μg/m$^3$), and the average value of EC in the upper reaches of the Shiyang River was 90.5 μs/cm from November 6 to 7 2016. During the dusty weather in Wuwei City, the AQI from November 24 to 26 in 2016 was moderately polluted (the AQI reached 166), and reached a level of heavy pollution on November 25 (PM2.5 was 114 μg/m$^3$ and PM10 was 396 μg/m$^3$), with the average value of EC in the upper reaches of the Shiyang River Basin at 215.55 μs/cm during the same period. Although the difference in the precipitation amount was small between the two precipitation processes, the average value of EC in the upper reaches of the Shiyang River in the later period was more than twice that of the former period. It can be seen that the dusty weather causes the solid particles in the air to increase, and the total dissolved solids in the precipitation to increase, so the value of EC in precipitation after sandstorm weather tended to be higher.

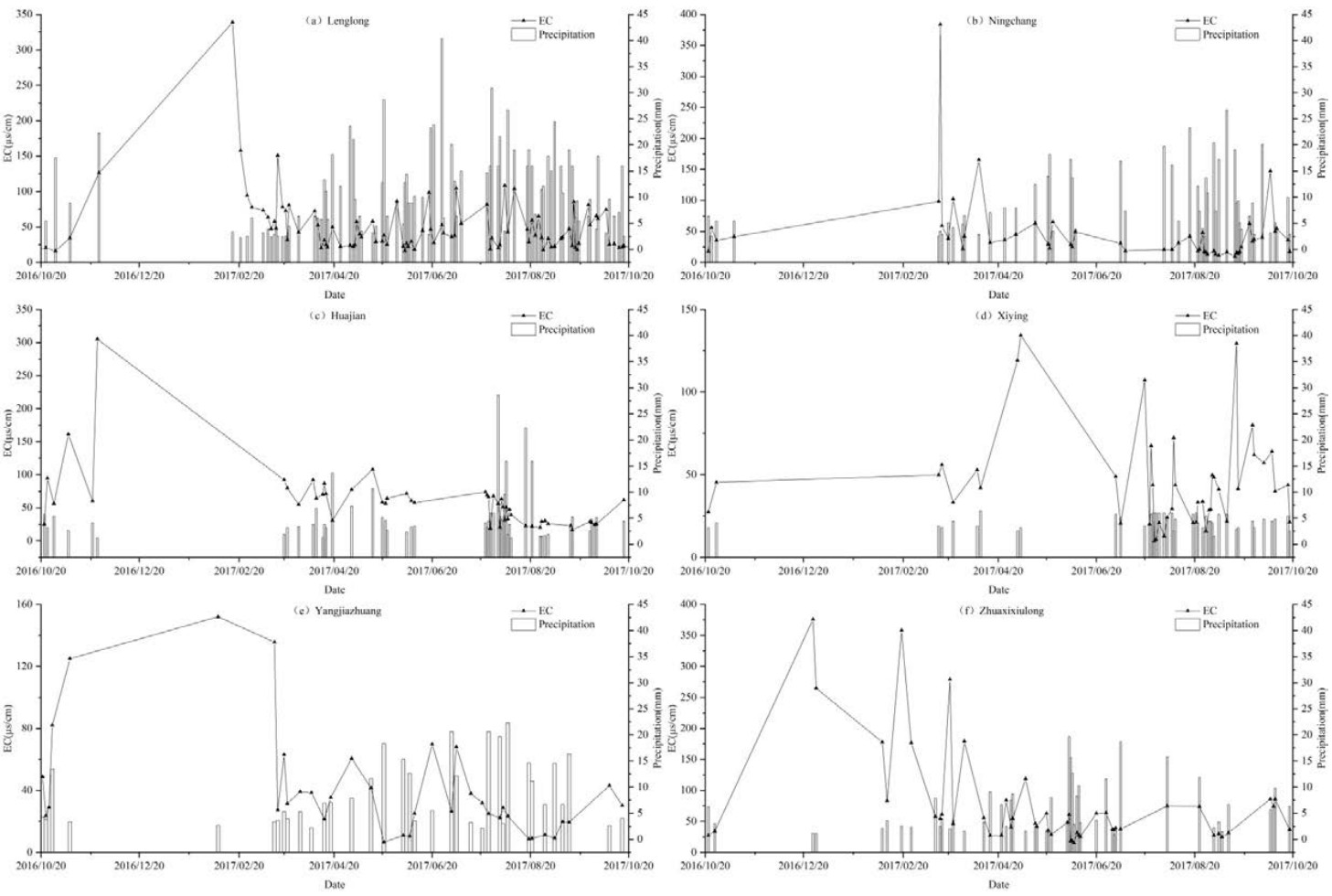

**Figure 4.** Seasonal variation in precipitation and electrical conductivity (EC).

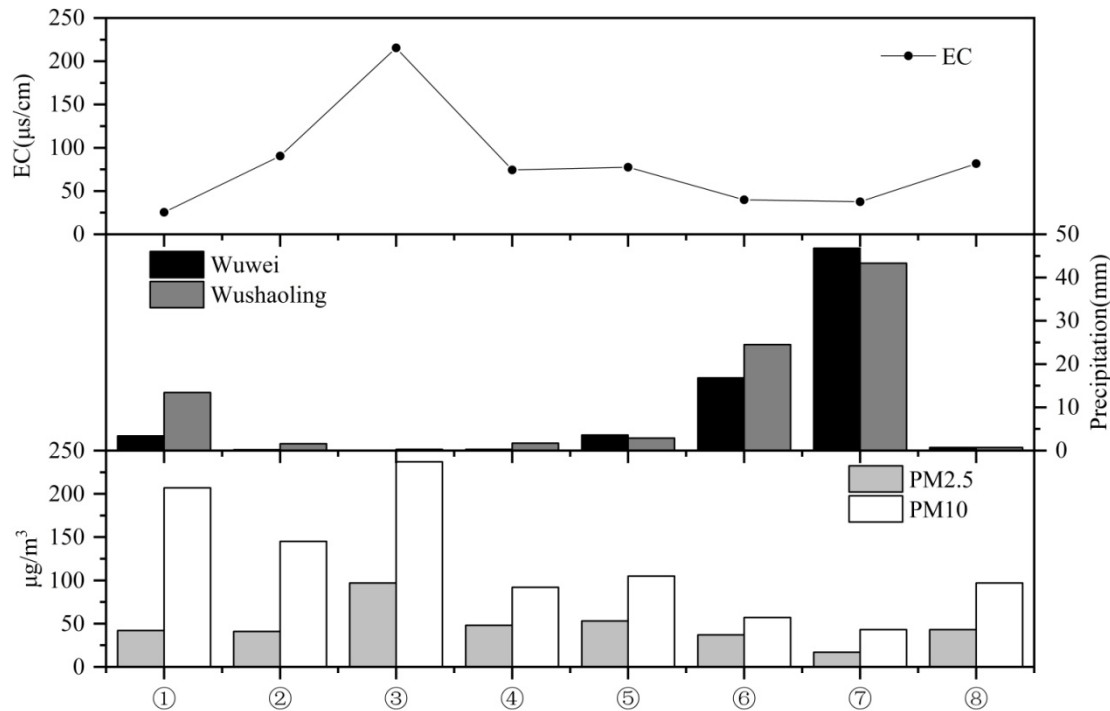

**Figure 5.** EC and pH of precipitation samples in dust weather.

**Table 3.** EC and pH of precipitation samples in dust weather.

| Example | Record Time | AQI | PM2.5 ($\mu g/m^3$) | PM10 ($\mu g/m^3$) | Precipitation Time | Wuwei (mm) | Wushaoling (mm) | EC ($\mu s/cm$) | pH |
|---|---|---|---|---|---|---|---|---|---|
| ① | 19 October 2016–20 October 2016 | 129 | 42 | 207 | 21 October 2016–22 Octorber 2016 | 3.4 | 13.4 | 25.59 | 7.56 |
| ② | 5 November 2016–6 November 2016 | 98 | 41 | 145 | 6 November 2016–7 November 2016 | 0.2 | 1.6 | 90.5 | 7.88 |
| ③ | 24 November 2016–26 November 2016 | 166 | 97 | 237 | 25 November 2016 | 0.1 | 0.3 | 215.55 | 7.38 |
| ④ | 15 March 2017–17 March 2017 | 72 | 48 | 92 | 16 March 2017–17 March 2017 | 0.3 | 1.7 | 74.33 | 7.72 |
| ⑤ | 20 March 2017–21 March 2017 | 79 | 53 | 105 | 22 March 2017–23 March 2017 | 3.6 | 2.9 | 77.6 | 7.68 |
| ⑥ | 3 June 2017–4 June 2017 | 65 | 37 | 57 | 4 June 2017–5 June 2017 | 16.8 | 24.5 | 40.02 | 7.65 |
| ⑦ | 20 July 2017–22 July 2017 | 47 | 17 | 43 | 23 July 2017–27 July 2017 | 46.8 | 43.3 | 37.68 | 7.38 |
| ⑧ | 5 October 2017–7 October 2017 | 74 | 43 | 97 | 7 October 2017 | 0.7 | 0.7 | 81.83 | 7.02 |

Note: the standard for the classification calculation of the air quality index (AQI) is the new ambient air quality standard (GB3095-2012). AQI, PM2.5 and PM10 are the average values of the corresponding periods, and EC and pH are the VWM of the samples collected at each sampling point.

## 4.2. Neutralizing Factor of Cations

The study area is located at the edge of the Tengger and Badain Jaran Deserts. Affected by sand and dust weather, there are more solid particles in the atmosphere there. The upper reaches of the Shiyang River are mainly composed of metamorphic rock such as mica schist, quartzite and granite and the main chemical composition is $SiO_2$, K, Na, Ca and other alkali metals or alkaline earth metal aluminosilicate minerals [45–48]. Therefore, the solid particles in the atmosphere have a certain neutralization effect on acidic pollutants (Table 4).

**Table 4.** Neutralization factor (NF) of cations at different sampling points.

| Sampling Point | Na$^+$ | K$^+$ | Mg$^{2+}$ | Ca$^{2+}$ | NH$_4{}^+$ |
|---|---|---|---|---|---|
| Lenglong | 0.92 | 0.27 | 0.32 | 3.32 | 1.47 |
| Ningchang | 1.55 | 0.81 | 0.94 | 5.32 | 4.95 |
| Huajian | 0.81 | 0.33 | 0.67 | 2.87 | 16.75 |
| Xiying | 1.08 | 0.67 | 0.85 | 4.45 | 2.3 |
| Yangjiazhuang | 1.16 | 0.44 | 0.52 | 3.09 | 14.69 |
| Zhuaxixiulong | 1.33 | 0.73 | 0.87 | 3.75 | 2.52 |
| Qilian | 1.24 | 0.16 | 0.2 | 2.38 | 0.21 |

According to the correlation analysis of ions as tabulated in Table 5, it can be concluded that Na$^+$, Mg$^{2+}$ and Ca$^{2+}$ are highly correlated with SO$_4{}^{2-}$ and NO$_3{}^-$. This also confirms the correctness of the anion test.

**Table 5.** Pearson correlation analysis.

| Ion Type | Na$^+$ | NH$_4{}^+$ | K$^+$ | Mg$^{2+}$ | Ca$^{2+}$ | F$^-$ | Cl$^-$ | SO$_4{}^{2-}$ | NO$_3{}^-$ |
|---|---|---|---|---|---|---|---|---|---|
| Na$^+$ | 1 | | | | | | | | |
| NH$_4{}^+$ | 0.401 | 1 | | | | | | | |
| K$^+$ | 0.718 ** | 0.261 ** | 1 | | | | | | |
| Mg$^{2+}$ | 0.687 ** | 0.231 ** | 0.544 ** | 1 | | | | | |
| Ca$^{2+}$ | 0.735 ** | 0.402 ** | 0.501 ** | 0.669 ** | 1 | | | | |
| F$^-$ | 0.484 ** | 0.294 ** | 0.211 * | 0.408 ** | 0.522 ** | 1 | | | |
| Cl$^-$ | 0.948 ** | 0.391 ** | 0.833 ** | 0.599 ** | 0.675 ** | 0.456 ** | 1 | | |
| SO$_4{}^{2-}$ | 0.822 ** | 0.385 ** | 0.603 ** | 0.862 ** | 0.839 ** | 0.546 ** | 0.739 ** | 1 | |
| NO$_3{}^-$ | 0.597 ** | 0.471 ** | 0.463 ** | 0.481 ** | 0.708 ** | 0.397 ** | 0.616 ** | 0.597 ** | 1 |

Note: ** indicates significant correlation at level 0.01 (two-tailed) and * indicates significant correlation at level 0.05 (two-tailed).

Within the study area, NF$_{Ca}{}^{2+}$ > NF$_{Na}{}^+$ > NF$_{Mg}{}^{2+}$ (Figure 6), due to the composition of the parent rock and the nature of the Ca element. There is a large amount of CaCO$_3$ in rock debris and dust after weathering, and this reacts with weak acids (HNO$_3$, H$_2$SO$_4$ and others) in rainwater to form neutral alkali metal salts, and neutralizes the acidity of precipitation in the region. The result is that the precipitation in this area was basically slightly alkaline throughout the year. The source of NH$_4{}^+$ is mainly related to human farming activities, NF$_{NH4}{}^+$ is often higher in populated village areas (Huajian and Yangjiazhuang), which indicates that the pH of precipitation at this sampling point was significantly affected by human factors.

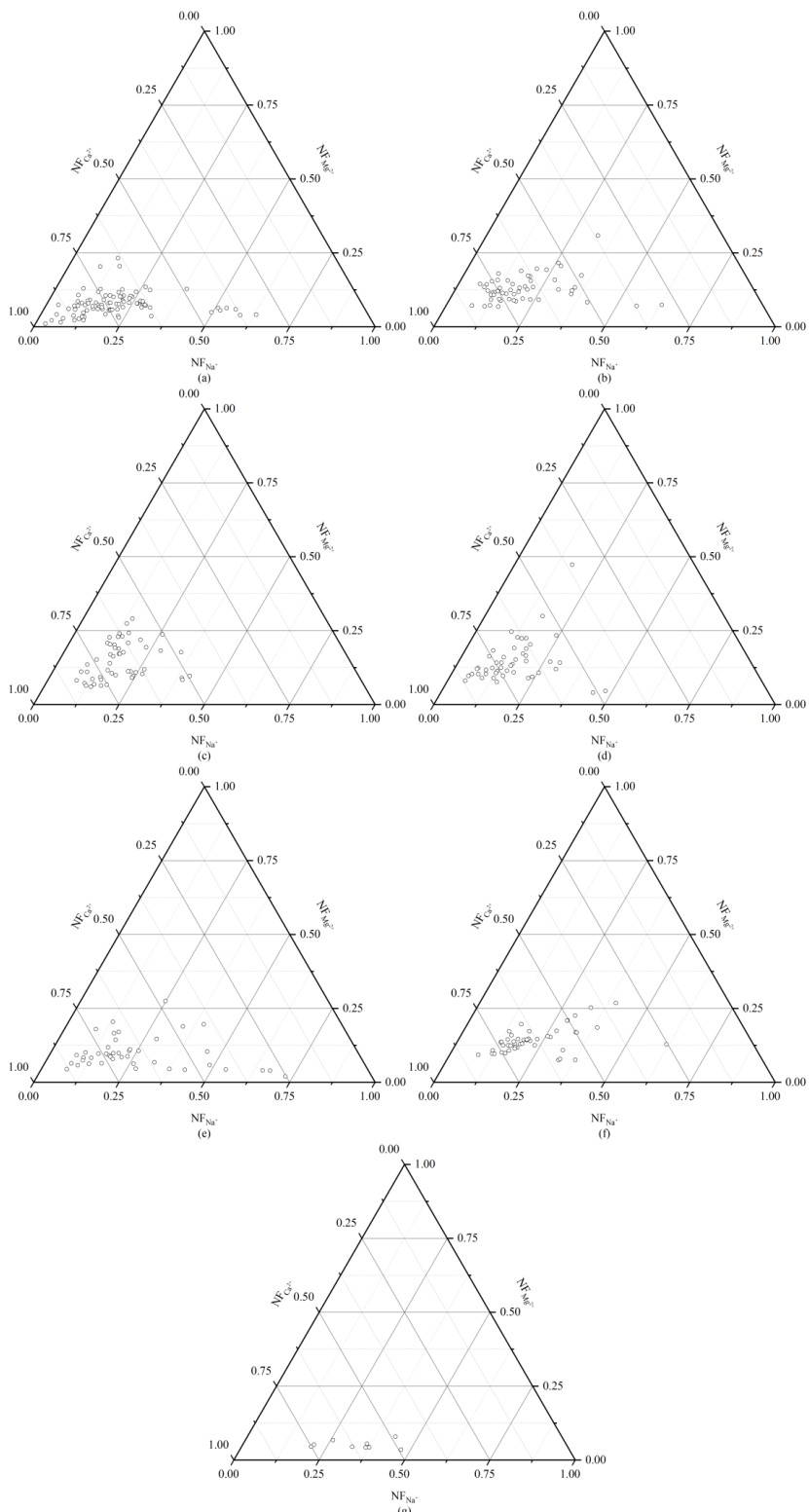

**Figure 6.** Triangular diagrams of NF for Na$^+$, Mg$^{2+}$ and Ca$^{2+}$ (**a**) Lenglong, (**b**) Ningchang, (**c**) Huajian, (**d**) Xiying, (**e**) Yangjiazhuang, (**f**) Zhuaxixiulong, (**g**) Qilian.

*4.3. Seasonal Variation of Ion Concentration in Precipitation*

Except for the ion NH$_4$$^+$, the change trend of anion and cation concentrations in precipitation is basically the same in the upper reaches of the Shiyang River (Figure 7). After entering the dry season in October, precipitation increased gradually, and reached its maximum in winter. However, at the

onset of the wet season in April each successive year when precipitation increased gradually and the ion concentration became diluted, the concentrations of various ions decreased and reached their minimum in summer.

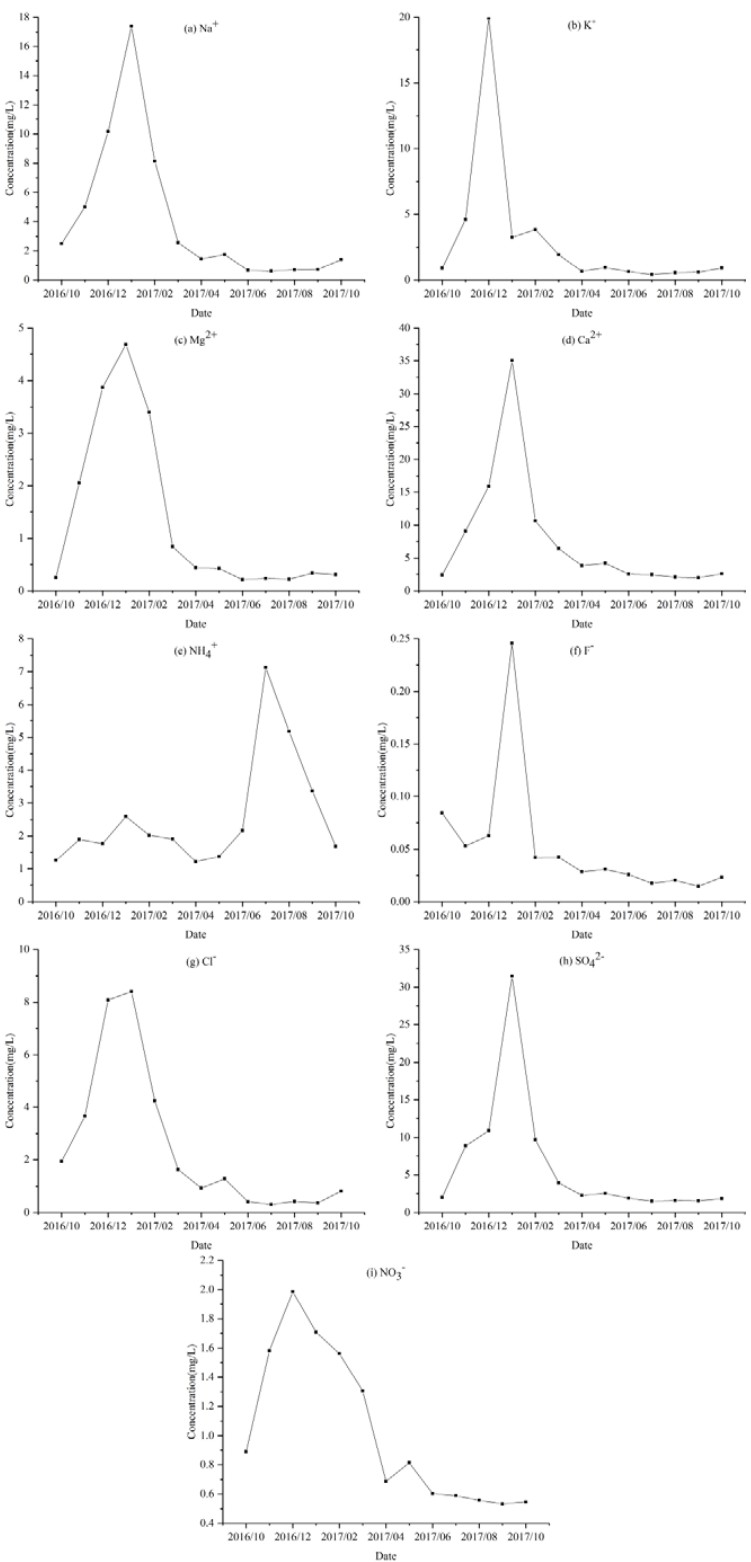

**Figure 7.** Seasonal variation of ion concentration in the study area (calculate the volume-weighted average monthly).

The maximum concentration of $NH_4^+$ occurs in summer, and it is mainly affected by human factors. The soil fertility is lower in the study area where the soil types are mostly gray brown desert soil, sandy soil, meadow soil, irrigated desert soil and ash calcium soil. Due to the lack of nitrogen in the area, it is often necessary to supplement nitrogen compounds, such as $NH_4HCO_3$ and $NH_4NO_3$.However, since the ammonium salt in the fertilizer is decomposed easily by heat, this causes the content of $NH_3$ to increase in the air and leads the ion concentration of $NH_4^+$ in precipitation to rise to its highest value in summer. The higher concentration of ions in the precipitation in the dry season is due to less precipitation events and precipitation in the winter and spring, lower relative humidity and more aerosol particles in the atmosphere [49–51]. However, in the wet season, the frequency and amount of precipitation was higher relatively, and the ion concentration diluted, which reduced the ion concentration in precipitation.

In addition, the ion concentration in precipitation was also affected by dusty weather, especially in spring when this weather occurred frequently. According to air quality data of Wuwei City, moderate pollution (AQI was 183), severe pollution (AQI was 258) and severe pollution (AQI was 391) occurred on 3, 4 and 5 May in 2017 respectively, and severe pollution (AQI was 361), moderate pollution (AQI was 162) and mild pollution (AQI was 127) occurred on 29, 30 and 31 May in 2017 respectively. Due to the dusty weather, the ion concentration in precipitation during May increased in the upper reaches of the Shiyang River.

### 4.4. Wet Deposition of Nitrogen (N) and Sulfur (S)

Wet deposition is an important form of N and S from the atmosphere. The sources of N include nitrogen oxide from lightning, $NO_X$ from automobile exhaust and ammonium fertilizer used in agriculture. The main source of S is the combustion of fossil fuels containing sulfur and the smelting of nonferrous metals. Nitrogen oxides such as $NO_2$ and $SO_2$ in the atmosphere can further react with water vapor to form $HNO_3$ and $H_2SO_4$, and then affect the natural environment. At the same time, under the influence of human activities, the wet deposition brings about excessive element input, and as such directly affects the material circulation and energy flow of the ecosystem, such as the composition of surface materials and water eutrophication. Therefore, the calculation of wet deposition of nitrogen and sulfur is helpful to study the composition of precipitation in this region.

From the Tables 6 and 7, we can see the change in wet deposition, in the research area under discussion, was greatly affected by precipitation and the frequency of precipitation. In the upper Xiying River area (Lenglong, Ningchang, Huajian and Xiying), the precipitation followed the vertical zonal differentiation law: as the altitude increased, the precipitation gradually increased and the wet deposition of N and S gradually increased. Huajian, Yangjiazhuang and Zhuaxixiulong were all gathering points for residents, and therefore were greatly affected by human activities, resulting in the increase of compounds containing N and S in the atmosphere. This was especially the case regarding ammonium salt fertilizers used during the agricultural activity period, coal heating in winter and automobile exhaust emissions all year round. A small coal mine, no longer in use, is located near the sampling site of Lenglong, and some of the coal mines are still kept and used in the factory area after mining. Therefore, the relatively high content of compounds containing S in the air led to large wet deposition of S. In terms of seasonal variation, the settlement flux of the wet season at each sampling point was basically greater than during the dry season, and the wet settlement reached its highest level in summer but tended to be lower in winter. The reason why wet deposition of N was higher in summer might be due to the thermal decomposition of fertilizer and $NH_3$ volatilization in farmland.

**Table 6.** The wet deposition flux of the nitrogen element (mg/dm$^2$).

| Sampling Point | Precipitation (mm) | Amount | Dry Season | Wet Season | Spring | Summer | Autumn | Winter |
|---|---|---|---|---|---|---|---|---|
| Lenglong | 1025.12 | 16.74 | 7.69 | 9.04 | 4.94 | 3.68 | 5.39 | 2.73 |
| Ningchang | 469.44 | 6.6 | 2.83 | 3.77 | 1.05 | 2.99 | 1.27 | 1.29 |
| Huajian | 282.05 | 4.78 | 1.38 | 3.4 | 2.36 | 1.41 | 1.01 | |
| Xiying | 197.67 | 2.95 | 0.82 | 2.95 | 0.73 | 2.35 | 0.67 | 0.02 |
| Yangjiazhuang | 613.54 | 7.62 | 1.98 | 5.65 | 2.15 | 3.2 | 2.27 | |
| Zhuaxixiulong | 290.63 | 6.72 | 3.4 | 3.32 | 1.86 | 2 | 0.79 | 2.07 |

**Table 7.** The wet deposition flux of the sulfur element (mg/dm$^2$).

| Sampling Point | Precipitation (mm) | Amount | Dry Season | Wet Season | Spring | Summer | Autumn | Winter |
|---|---|---|---|---|---|---|---|---|
| Lenglong | 1025.12 | 30.73 | 9.92 | 20.82 | 5.31 | 13.44 | 6.96 | 5.02 |
| Ningchang | 469.44 | 4.79 | 2.02 | 2.57 | 1.33 | 1.68 | 1.18 | 0.6 |
| Huajian | 282.05 | 5.95 | 1.58 | 4.37 | 2.87 | 1.67 | 1.41 | |
| Xiying | 197.67 | 3.54 | 0.77 | 2.77 | 0.32 | 2.26 | 0.64 | 0.02 |
| Yangjiazhuang | 613.54 | 7.38 | 1.35 | 6.02 | 2.35 | 3.22 | 1.8 | |
| Zhuaxixiulong | 290.63 | 6.31 | 2.97 | 3.34 | 1.58 | 2.29 | 1.05 | 1.4 |

## 5. Discussion

### 5.1. Relationship between EC and Elevation

Studies have shown that sand dust distributes mainly into atmosphere to a height of between 1 and 4 km and that the vertical distribution of mass concentration of sand dust ($\mu$g/m$^3$) in the atmosphere appeared parabolic. Furthermore, it reaches its highest value at about 1500 m above the ground [46]. As the altitude increased, the atmospheric aerosol settled further and the mass concentration of sand dust in atmosphere decreased gradually. In the Xiying River valley in the upper reaches of the Shiyang River, four sampling points Xiying, Huajian, Ningchang and Lenglong were arranged along altitude laterality (Figure 1), and the distance between each sampling point was about 20 km. The VWM of EC at the Xiying, Huajian, Ningchang and Lenglong sampling point were 48.93 $\mu$s/cm, 58.67 $\mu$s/cm, 47.32 $\mu$s/cm and 46.48 $\mu$s/cm respectively, which indicates that the conductivity of precipitation decreased gradually along with the rise in altitude (Figure 8). However, the EC at HJ sampling point was higher than at the other three sampling points. Since this sampling point was located at the confluence of the Ningchang River and the Shuiguan River, where the residential areas concentrate relatively close to the provincial road. Due to human factors, there were more particles in the air, and the EC in precipitation was also higher. This situation was also evident at the sampling points of Yangjiazhuang, Qilian and Zhuaxixiulong. However, the EC in the precipitation at sampling point Zhuaxixiulong, which sits at a higher altitude than the other six sampling points, was also affected by human factors, being situated close to the Lianhuo and Xugu highways, and the residential areas also being concentrated relatively nearby. Since daily life of the residents and a prevalence of motor vehicles generate much more dust and particles into the atmosphere, the EC in the precipitation in this region was higher than in the other regions under discussion.

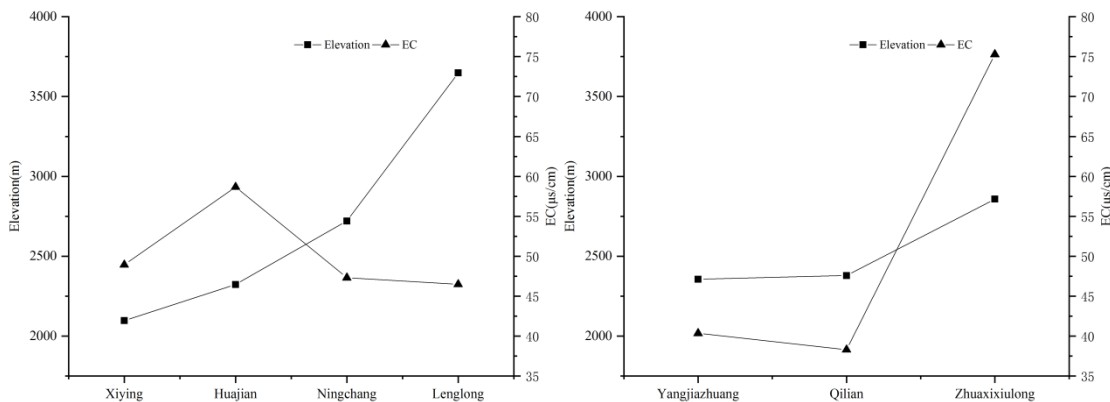

**Figure 8.** Changes of EC in the precipitation along with altitude in the upper reaches of the Shiyang River.

*5.2. Composition and Comparison of Precipitation Ions*

In the upper reaches of Shiyang the River, the VWM of $Na^+$, $K^+$, $Mg^{2+}$, $Ca^{2+}$ and $NH_4^+$ in precipitation were 1.74, 1.28, 0.57, 3.78 and 3.10 mg/L, and the VWM of anions concentrations of $F^-$, $Cl^-$, $NO_3^-$ and $SO_4^{2-}$ were 0.04, 1.10, 0.78 and 2.72 mg/L. The order of cation concentrations in the precipitation was $Ca^{2+} > NH_4^+ > Na^+ > K^+ > Mg^{2+}$ (Figure 9), which is the same as the order of the standard crustal mineral concentrations ($Ca^{2+} > Na^+ > K^+ > Mg^{2+}$). However, the order of anion concentrations in the precipitation was $SO_4^{2-} > Cl^- > NO_3^- > F^-$ (Figure 9), which is different from the order of anion concentrations contained in standard seawater ($Cl^- > SO_4^{2-} > NO_3^-$), and the concentration of $SO_4^{2-}$ was higher than that of $Cl^-$ [52–58]. The main ions in precipitation were $Ca^{2+}$, $NH_4^+$, $Na^+$, $Cl^-$ and $SO_4^{2-}$, which accounted for 82.33% of the total cation concentration and 99.14% of the total anion concentration. $Ca^{2+}$ was the first dominant cation that accounted for 34.37% of the total cation concentration, and $SO_4^{2-}$ was the first dominant anion that accounted for 57.76% of the total anion concentration. So, the type of precipitation in the upper reaches of the Shiyang River was the $SO_4^{2-}$–$Ca^{2+}$ type, which is consistent with the type of precipitation in the Yulong Mountain Baishui Glacier No.1 during the wet season [59,60]. The ion concentrations of $NH_4^+$ atthe sampling points Huajian and Yangjiazhuang were higher relative to the other sampling point sites. This is because these two sampling points are close to residential areas, and the fertilizer is used widely in agricultural production. The ammonium salt of the fertilizer had poor thermal stability and decomposes easily with heat, and $NH_3$ diffused into the atmosphere, dissolving easily in water and eventually forming $NH_4^+$ (Table 8).

As shown in Table 8, the precipitation in the upper reaches of the Shiyang River Basin was compared with that in the other areas. Now we will consider these comparisons.

Compared with the Waliguan station and the Central Qinghai–Tibet Plateau. The ion concentrations ofNa$^+$, $K^+$, $Mg^{2+}$, $Ca^{2+}$, $NH_4^+$, $Cl^-$, $NO_3^-$ and $SO_4^{2-}$ in precipitation in the upper reaches of Shiyang river were 3.38, 1.29, 2.48, 1.57, 4.48, 1.11, 0.81 and 1.13 times as high as those in Waliguan, and 4.43, 2.17, 5.78, 0.59, 1.54, 1.25 and 3.42 times as high as those in Central Qinghai–Tibet Plateau (vacancy of $NH_4^+$). The study area is close to the Tengger Desert, and is frequently affected by dusty weather, precipitation events and precipitation are less, therefore the ion concentrations were higher relatively. While the altitudes of the Waliguan and the Central Qinghai–Tibet Plateau were higher than the study area, the atmospheric aerosol was affected by natural sedimentation, which caused the ion concentration to lower.

Compared with the Yushugou area of the Tianshan Mountains. The ion concentrations of $Na^+$, $K^+$, $Mg^{2+}$, $Ca^{2+}$, $NH_4^+$, $Cl^-$, $NO_3^-$ and $SO_4^{2-}$ in precipitation in the upper reaches of Shiyang River were 0.14, 0.36, 0.50, 1.33, 0.43, 0.33, 0.05, 0.11 and 0.56 times as high as those in the Yushugou area of Tianshan Mountain. The drought index of the study area was lower, and thus correspondingly the ion concentration in precipitation was also lower. In addition, the ion concentrations of $NO_3^-$ and $SO_4^{2-}$ were relatively higher in the Yushugou area because of the location where it is close to Hami

City, and the ion concentrations of $Ca^{2+}$ was relatively higher in the study area because of the rock composition there.

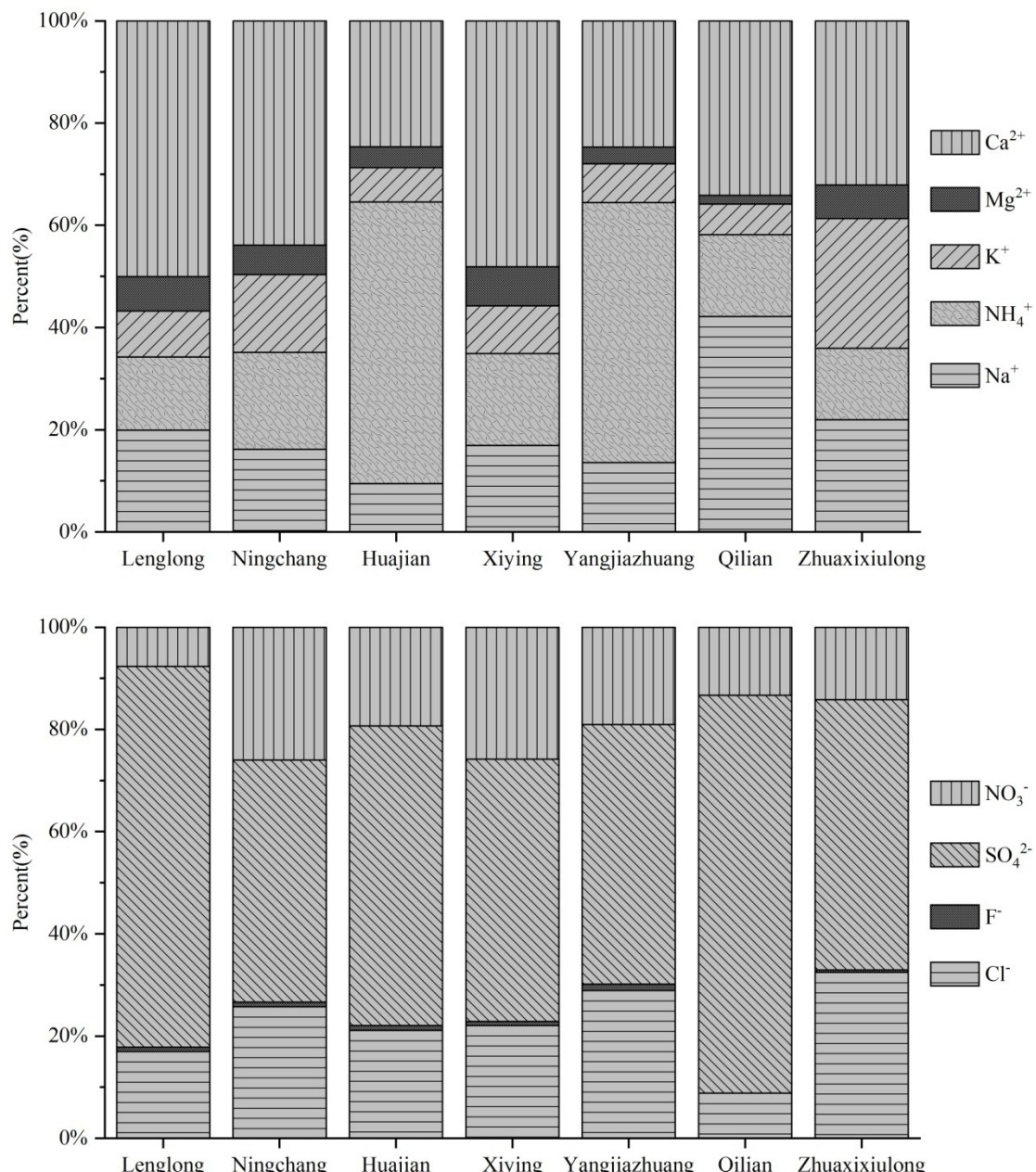

**Figure 9.** Percentage accumulation diagram of the ion concentration in the precipitation.

Compared with Lanzhou and Xi'an. The ion concentrations of $Na^+$, $K^+$, $Mg^{2+}$, $Ca^{2+}$, $NH_4^+$, $Cl^-$, $NO_3^-$ and $SO_4^{2-}$ in precipitation in the upper reaches of Shiyang River were 6.65, 4.41, 0.93, 0.20, 3.04, 0.15, 1.06, 0.17 and 0.25 times as high as those in Lanzhou City, and 2.59, 2.29, 1.18, 0.42, 0.76, 0.77, 0.10 and 0.11 times as high as those in Xi'an City. The ion concentrations of $Na^+$ and $K^+$ in precipitation in Lanzhou and Xi'an were less than those in the study area, but the ion concentrations of $F^-$, $NO_3^-$, $SO_4^{2-}$ and $Ca^{2+}$ indicated the opposite. Due to pollutants and solid particles coming from industry, traffic and construction, the input ratios of $F^-$, $NO_3^-$, $SO_4^{2-}$ and $Ca^{2+}$ were larger in these two cities. However, this was not the case in the upper reaches of the Shiyang River Basin, where the overall effect of human activities was lower.

**Table 8.** Main ion concentrations of precipitation in this research and comparison with other regions.

| Sampling Station | Elevation | Na$^+$ | K$^+$ | Mg$^{2+}$ | Ca$^{2+}$ | NH$_4^+$ | F$^-$ | Cl$^-$ | NO$_3^-$ | SO$_4^{2-}$ | Literature Sources |
|---|---|---|---|---|---|---|---|---|---|---|---|
| Tianshan Mountain | 1670 | 2.67 | 1.08 | 0.30 | 3.48 | 2.11 | 0.06 | 2.71 | 2.74 | 4.83 | Ref. [44] |
| Waliguan | 3816 | 0.55 | 0.96 | 0.21 | 2.26 | 0.70 | n.d. | 0.93 | 0.95 | 2.24 | Ref. [61] |
| Qinghai–Tibet Plateau | 4730 | 0.42 | 0.57 | 0.09 | 6.02 | n.d. | n.d. | 0.67 | 0.62 | 0.74 | Ref. [62] |
| Lanzhou | 1513 | 0.28 | 0.28 | 0.56 | 17.72 | 1.03 | 0.26 | 0.98 | 4.61 | 9.98 | Ref. [63] |
| Xi'an | 408 | 0.72 | 0.54 | 0.44 | 8.51 | 4.12 | 0.55 | 1.35 | 7.99 | 23.51 | Ref. [64] |
| Wuwei | 1532 | 3.90 | 1.70 | 1.60 | 19.40 | 1.60 | 0.10 | 4.20 | 7.10 | 23.10 | Ref. [65] |
| Minqin | 1368 | 6.20 | 2.70 | 1.70 | 13.50 | 4.40 | 0.10 | 12.30 | 13.90 | 12.10 | Ref. [65] |
| Lenglong | 3648 | 1.31 | 0.61 | 0.61 | 6.43 | 1.00 | 0.01 | 0.71 | 2.97 | 0.31 | This research |
| Ningchang | 2721 | 0.90 | 0.79 | 0.64 | 5.06 | 1.29 | 0.01 | 0.63 | 1.03 | 0.55 | This research |
| Huajian | 2323 | 1.04 | 0.78 | 0.95 | 6.02 | 10.75 | 0.01 | 0.79 | 2.16 | 0.70 | This research |
| Xiying | 2097 | 0.98 | 0.59 | 0.90 | 5.82 | 1.32 | 0.01 | 0.71 | 1.79 | 0.92 | This research |
| Yangjiazhuang | 2356 | 0.67 | 0.49 | 0.33 | 3.60 | 8.20 | 0.01 | 0.56 | 1.32 | 0.65 | This research |
| Zhuaxixiulong | 2858 | 1.59 | 1.58 | 1.00 | 6.68 | 1.75 | 0.01 | 1.03 | 2.17 | 0.72 | This research |
| Qilian | 2379 | 2.55 | 0.36 | 0.10 | 2.07 | 0.97 | n.d. | 0.22 | 0.34 | 1.97 | This research |

Note: n.d. indicates it is not detected. The elevation unit is m. The literature data are converted equivalent units (µeq/L) into concentration units (mg/L) with two decimal places.

Compared with Wuwei and Minqin. The ion concentrations of $Na^+$, $K^+$, $Mg^{2+}$, $Ca^{2+}$, $NH_4^+$, $Cl^-$, $NO_3^-$ and $SO_4^{2-}$ in precipitation in the upper reaches of Shiyang River were 0.48, 0.73, 0.33, 0.18, 1.96, 0.38, 0.25, 0.11 and 0.11 times as high as those in Wuwei City, and 0.3, 0.46, 0.31, 0.26, 0.71, 0.08, 0.06 and 0.21 times an high as those in Minqin county. It can be seen that the ion concentrations in precipitation of the Shiyang River Basin increased gradually from upstream to downstream, explained primarily because of human factors and dusty weather. The oasis located in the middle and lower reaches of the Shiyang River Basin has active economic activities, and is located close to the Tengger Desert, so the ion concentrations in precipitation increased relatively in these areas.

## 5.3. Source of Ions in Precipitation

In the upper reaches of the Shiyang River, the seawater input of $Ca^{2+}$ and the land-source input of $Cl^-$ in precipitation were small (Tables 9 and 10), which also confirms the correct selection of using $Ca^{2+}$ as a terrestrial reference source and using $Cl^-$ as a sea salt source indicator. It can be considered that the ion of $Cl^-$ was from the sea source input, and that the ion of $Ca^{2+}$ was from the land source input in the study area. There were two sources for the ion of $Na^+$, one being the sea source input (about 35%), the other the land source input (about 65%). The sea source comes from oceanic air masses travelling long-distance while the land source comes from salt particles on the surface of the earth's crust that enter into the atmosphere by means of wind erosion and then dissolves into precipitation. In the study area, the soil salinization situation was more serious. The average salinity was about 0.025 g/L. Similar to the ion of $Ca^{2+}$, the main sources of $K^+$ and $Mg^{2+}$ wereland-source input, which is mainly related to rock weathering, but there was also a small amount of sea source input.

**Table 9.** Seasonal variation of enrichment factor (EF) of the main ions.

| Source | Ion | Spring | Summer | Autumn | Winter | Dry Season | Wet Season |
|--------|-----|--------|--------|--------|--------|------------|------------|
| $EF_{sea}$ | $Na^+$ | 2.71 | 3.11 | 2.54 | 2.99 | 2.94 | 2.86 |
| | $K^+$ | 44.25 | 67.52 | 49.33 | 62.12 | 58.87 | 49.08 |
| | $Mg^{2+}$ | 6.76 | 8.9 | 6.6 | 8.71 | 8.13 | 7.63 |
| | $Ca^{2+}$ | 176.18 | 291.48 | 110.65 | 135.28 | 133.26 | 215.06 |
| | $SO_4^{2-}$ | 16.24 | 31.42 | 14.99 | 17.76 | 16.99 | 21.86 |
| | $NO_3^-$ | 24,667 | 52,097.5 | 17,620.5 | 9437.4 | 11,207.37 | 34,431.87 |
| $EF_{crust}$ | $Na^+$ | 0.61 | 0.42 | 0.91 | 0.88 | 0.88 | 0.53 |
| | $K^+$ | 0.25 | 0.23 | 0.45 | 0.46 | 0.44 | 0.23 |
| | $Mg^{2+}$ | 0.35 | 0.28 | 0.55 | 0.59 | 0.56 | 0.33 |
| | $Cl^-$ | 48.18 | 29.12 | 76.72 | 62.75 | 63.7 | 39.47 |
| | $SO_4^{2-}$ | 12.62 | 14.75 | 18.53 | 17.97 | 17.44 | 13.91 |
| | $NO_3^-$ | 29.83 | 38.08 | 33.92 | 14.86 | 17.92 | 34.11 |

The Qilian Mountain is mainly composed of metamorphic rocks, such as mica schist, quartzite and granite [25], so their main chemical components include alkali metal or alkaline earth metal elements such as K, Na and Ca. Therefore, the ions of $Ca^{2+}$, $K^+$ and $Mg^{2+}$ enter into the atmosphere with the weathering of this rock, and then enter into the precipitation. The ions of $SO_4^{2-}$ and $NO_3^-$ come from human activity inputs, mainly in the form of burning of fossil fuels and the emission of automobile exhaust fumes. In the upper reaches of the Shiyang River, there are many rural residential areas where coal is the main fuel used. There are also several national and provincial roads where the traffic volumes are higher and these then become significant sources of $SO_4^{2-}$ and $NO_3^-$. The vast majority of $NH_4^+$ also comes from human activities primarily in the form of fertilizers used in agricultural activities, while a little comes from the decomposition of microorganisms. Large arable land is found distributed along the river valley in the upper reaches of the Shiyang River, and nitrogen fertilizer is used widely in agricultural production. So, the decomposition of ammonium salt becomes the main source of $NH_4^+$ in precipitation.

**Table 10.** Seasonal variation of the contribution rate of the main ion sources.

| Source | Ion | Spring | Summer | Autumn | Winter | Dry Season | Wet Season |
|--------|-----|--------|--------|--------|--------|-----------|-----------|
| SSF | $Na^+$ | 36.90% | 32.13% | 39.31% | 33.41% | 34.00% | 34.99% |
| | $K^+$ | 2.26% | 1.48% | 2.03% | 1.61% | 1.70% | 2.04% |
| | $Mg^{2+}$ | 14.80% | 11.23% | 15.15% | 11.48% | 12.31% | 13.10% |
| | $Ca^{2+}$ | 0.57% | 0.34% | 0.90% | 0.74% | 0.75% | 0.46% |
| | $SO_4^{2-}$ | 6.16% | 3.18% | 6.67% | 5.63% | 5.89% | 4.58% |
| | $NO_3^-$ | 0.00% | 0.00% | 0.01% | 0.01% | 0.01% | 0.00% |
| CF | $Na^+$ | 63.10% | 67.87% | 60.69% | 66.59% | 66.00% | 65.01% |
| | $K^+$ | 97.74% | 98.52% | 97.97% | 98.39% | 98.30% | 97.96% |
| | $Mg^{2+}$ | 85.20% | 88.77% | 84.85% | 88.52% | 87.69% | 86.90% |
| | $Cl^-$ | 2.08% | 3.43% | 1.30% | 1.59% | 1.57% | 2.53% |
| | $SO_4^{2-}$ | 7.93% | 6.78% | 5.40% | 5.57% | 5.73% | 7.19% |
| | $NO_3^-$ | 3.35% | 2.63% | 2.95% | 6.73% | 5.58% | 2.93% |
| ASF | $SO_4^{2-}$ | 85.92% | 90.04% | 87.93% | 88.80% | 88.38% | 88.23% |
| | $NO_3^-$ | 96.64% | 97.37% | 97.05% | 93.26% | 94.41% | 97.07% |

From a seasonal variation perspective, the sources of $Na^+$, $K^+$ and $Mg^{2+}$ were similar. The sea source input in spring and autumn was slightly larger than that in summer and winter, but the opposite was true for the land source input. For the ions of $SO_4^{2-}$ and $NO_3^-$, the ratio of anthropogenic source input was the highest in summer, and the lowest rate of anthropogenic sources input appeared in spring for the ion of $SO_4^{2-}$, but this appeared in winter for the ion of $NO_3^-$. The ratios of the sea source input of $Na^+$, $K^+$ and $Mg^{2+}$ were larger in the wet season than that in dry season, while the ratios of anthropogenic source input were larger in the dry season than that during the wet season. For the ion $SO_4^{2-}$, the ratio of anthropogenic source input was larger in the dry season than that in the wet season, but this was converse for the ion $NO_3^-$. Though the ion sources in precipitation were slightly different in different seasons, the ion $Cl^-$ comes from a sea source while and the ions of $Na^+$, $K^+$, $Mg^{2+}$ and $Ca^{2+}$ mainly came from the land source, and the ions of $NH_4^+$, $SO_4^{2-}$ and $NO_3^-$ mainly from anthropogenic sources.

Using the HYSPLIT-4 model from the American Atmospheric Laboratory where an air mass is tracked before each precipitation, the black star represents the source, which is located at 37.83° N, 120.01° E. The source of water vapor was obtained by the backward trajectory method. In the upper reaches of the Shiyang River, the main sources of water vapor came from the westerly circulation system, the northern cold anticyclone, the East Asian monsoon, the South Asian monsoon and the thermal effects of the Qinghai–Tibet Plateau (Figure 10). The main sources of water vapor in the dry season were from the westerly circulation system and the northern cold anticyclone, but these form the westerly circulation and the monsoon circulation in the wet season. In summer when precipitation is concentrated, the main sources of water vapor were the East Asian and South Asian monsoons. From the path of water vapor transportation, it can be seen that the land sources input in precipitation in the upper reaches of the Shiyang River mainly came from the surface materials of the Xinjiang Province of China, Central Asia and the northern section of the Mongolian Plateau during the dry season, while the sea sources input mainly came from the ocean water vapor brought by the westerly, and monsoon circulation systems.

The air mass brought by the summer monsoon has high humidity and a lot of precipitation, while the winter westerly wind brings a lot of dust on the ground in Central Asia, resulting in an increase in aerosol particles. Therefore, the EC of precipitation is often lower in summer than in winter. In the spring, due to the dust caused by the prevailing westerly wind, the air contains more alkaline earth metal ions ($Mg^{2+}$, $Ca^{2+}$, etc.), which also leads to a higher pH and ion concentration in spring. Summer and autumn are affected by the warm and humid monsoon in East Asia, South Asia, more precipitation and heavy precipitation, diluting the pH and the main ion concentration. So, in different

seasons, different water vapor sources and their transport paths lead to differences in ion concentration in precipitation.

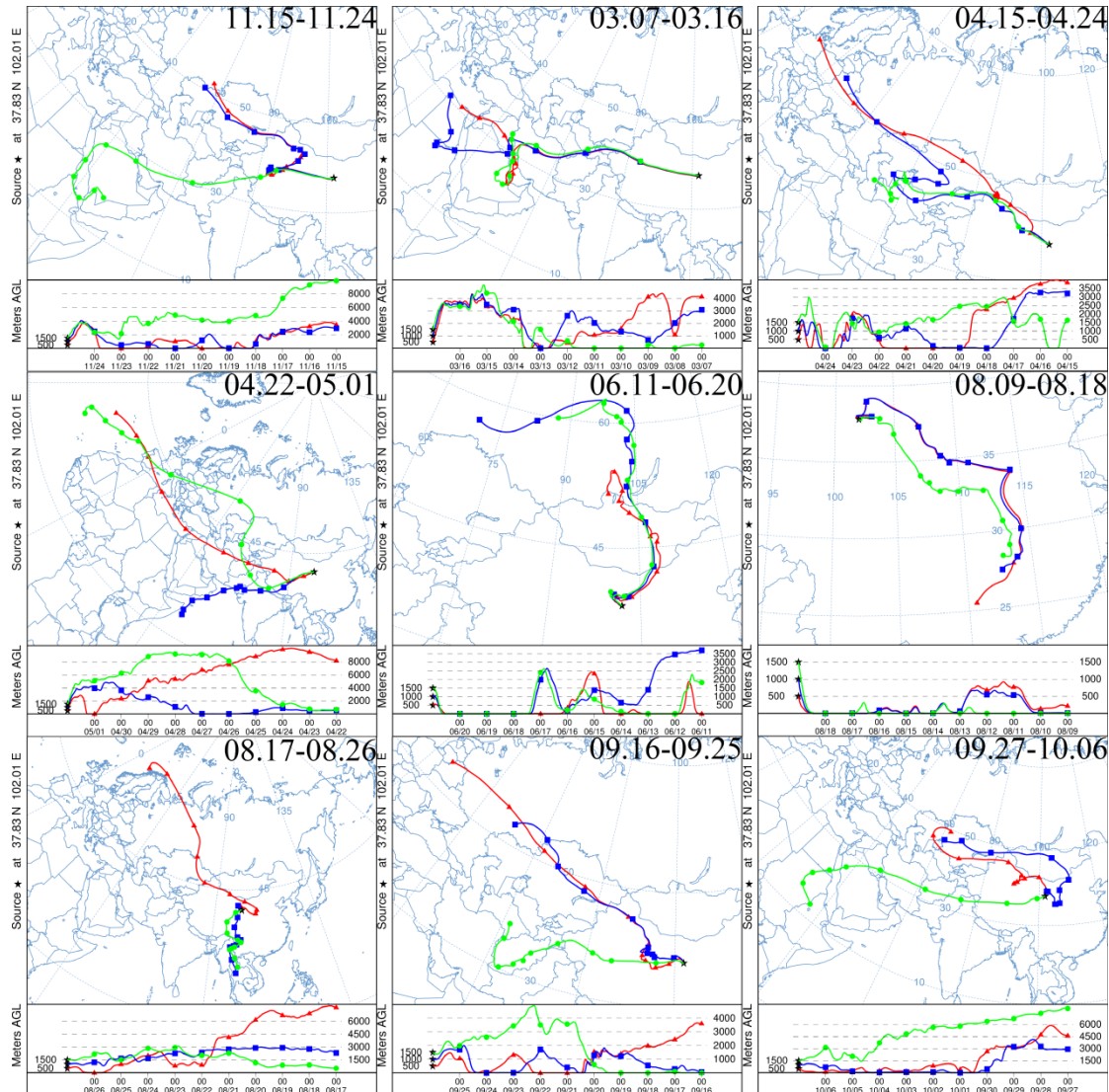

**Figure 10.** Backward trajectory of the water vapor source in the study area.

## 6. Conclusions

The precipitation in the upper reaches of the Shiyang River is mildly alkaline throughout the whole year, and the range of pH varied between 6.54 to 8.96. The cations with strong neutralization ability were $Ca^{2+}$ and $NH_4^+$. The order of VWM of pH in the different seasons was spring > winter > summer > autumn. The VWM of EC during the wet season was less than that during the dry season. With the increase in altitude, the EC of precipitation from downstream to upstream decreased gradually.

From upstream to downstream in the Shiyang River Basin, the ion concentration in the precipitation gradually increased. The concentration of cations in the upper reaches of the Shiyang River was the same as that of standard crustal minerals ($Ca^{2+}$ > $Na^+$ > $K^+$ > $Mg^{2+}$). The particles in the air of the study area were greatly affected by local terrestrial materials. The order of the anion concentration of precipitation was different from the order of the concentration of anions contained in standard seawater. The concentration of $SO_4^{2-}$ was higher than the concentration of $Cl^-$ due to human activities.

Influenced by precipitation and human factors, the amount of the wet deposition of N and S tended to reach its highest level in summer and the lowest in winter, and the wet deposition during the wet season was higher than that during the dry season.

West wind circulation systems and monsoon circulation systems were the main source of water vapor in the upper reaches of the Shiyang River. The main source of $Cl^-$ in precipitation was the ocean, $Ca^{2+}$, $K^+$ and $Mg^{2+}$ was from the earth's crust, $Na^+$ came from the ocean and earth crust, $NO_3^-$, $SO_4^{2-}$ and $NH_4^+$ came from human activities while $F^-$ came from nature and human activities and was rarely contained in precipitation.

**Author Contributions:** Data curation, W.J.; Investigation, X.M.; Software, X.X., R.Y., Y.S., L.Y., H.X.; Writing—original draft preparation, Z.Z.; Writing—review and editing, W.J., G.Z. All authors have read and agreed to the published version of the manuscript.

**Funding:** This research was funded by National Natural Science Foundation of China (41661005, 41867030, 41661084), National Natural Science Foundation innovation research group science foundation of China (41421061), and Autonomous project of State Key Laboratory of Cryosphere Sciences (SKLCS-ZZ-2017).

**Acknowledgments:** We would like to thank the colleagues in the Northwest Normal University for their help in writing process. We would like to thank to those who helped us in the experiment. We are grateful to anonymous reviewers and editorial staff for their constructive and helpful suggestions.

**Conflicts of Interest:** The authors declare no conflict of interest.

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
