# Peer review of "Hydrochemical Characteristics and Ion Sources of Precipitation in the Upper Reaches of the Shiyang River, China"

_water, doi:10.3390/w12051442_

Round 1
Reviewer 1 Report
1. General comments
The authors address an interesting subject in the context of the origin and nature of atmospheric contamination agents and the variation in the seasonal composition of atmos+heric precipitation. They also address the origin and paths of atmospheric air masses. The elements presented can serve as a standard for future analyzing the variation in the composition of interannual or anual atmospheric precipitation
This work is supported by a significant volume of data obtained, from 7 sampling sites, over a hydrological year and in a total of 355 samples. In this context, the authors must present a summary table with basic statistical parameters (minimum, maximum, average and standard deviation) for the different variables studied.he must presente.
Another table should show the temporal distribution (monthly) of the number of samples for each sampling point.
The authors report that the composition of fertilizers used in agriculture is very important for the NH4+ composition of atmospheric precipitation. Therefore, they should include an approach to the composition of the main fertilizers used in agricultural practices in the studied region.
2. Specific comments
Point 2. Study área, line 74 – The Authors should present a regional characterization of the geology and the mineralogical composition, if possible also present a geological map of the region. This is very important to explain the origin of some components in atmospheric precipitation.
Lines 89 to 93 - Replace “annual evaporation rate” with “potential annual evaporation rate” because in any situation the annual precipitation is less than the potential annual evaporation.
Lines 98 (figure 1), 219 (figure 3), 233 (figure 4), 279 (figure 5), 289 (figure 6) - improve readability.
Lines 138 and 139 - The presence of CO32- and HCO3- anions in atmospheric precipitation water is an assumption. The confirmation of the balance of ionic charges should be proved by presenting some results of the composition in these ions.
Line 164 – The relations Na+/Cl- e Mg2+/Cl- - what do the results represent? Annual average values? What is the annual range?
Line 200 - 4.Results - Are the pH, EC and C values used in the text, figures and tables in this chapter refer to the values calculated according to the formulas presented in lines 122, 123 and 124, respectively? This needs to be explained in the different contexts where these parameters are used.
Line 244 and table 2 (line 252) - Compare the PM 2.5 and PM 10 values of line 244 with the corresponding values in table 2 (bottom line of the table). Aren't they supposed to be the same?
Line 252, Table 2 - Suggestion: graphically represent the values of PM2.5 + PM10 vs EC and pH - it would be easier to visualize for interpretation.
Lines 257 to 261 - This characterization should be placed in point 2 (2. Study area), line 74, where this study is contextualized.
Lines 270 to 272 - This characterization element should be mentioned in point 2 (2. study area), line 74 .
Lines 357 to 361 - The impact of the human factor should be analyzed quantitatively and not qualitatively. Maybe apply the percentage ratio of (SO42- + NO3-) / EC.
Lines 426 to 428 - This content should be referred to in point 2 (2. Study area), page 74, with regard to the geological framework. The authors must also allude to the mineralogical composition of the granites to assess the importance of the contribution in the elements Na, Ca and K.
Lines 450 to 463 - The results of the HYSPLIT-4 application could be better explored by correlating the trajectories of atmospheric air masses with the chemical composition, pH and EC of atmospheric precipitation.
Line 465, figure 9 - in each figure put the month and year. Also indicate the “black star” as the study área.
Line 498 – (References) - put the year in the references:1, 3, 6, 8, 16, 19, 21, 23, 27, 31, 33, 35, 36, 46, 48, 51, 54, 55, 57, 58, 60, 61 and 62. Format references: 22, 32, 34 and 37.
Reviewer 2 Report
Very good work. Good luck.
Author Response
Thanks to the reviewers for taking your time to comment on my article
Reviewer 3 Report
This seems to be an interesting article about the ion content and hydrochemical characteristics of precipitation in the river basin of the Shiyang River. The topic seems to be novel, as there are no studies yet on the wet deposition of ions and elements on this river basin, and the data seems useful for further meteorological and ecological study.
I find the study to be scientifically consistent and, in general, most of the discussion and conclusions are well supported on the presented data.
However, I must advise the authors to seek English review of their manuscript. In its current form, it is understandable, but some instances the grammar could be quite improved and the text could become clearer and more correct.
The Figures (all of them, but in particular Figs.3, 4, 5, 6, 7 and 8) are very small and difficult to read and to interpret in detail. They must be enlarged for better understanding of the reader and adequately support the text.
I will point out my suggestions for the improvement of the manuscript before publication by section:
Abstract
- Line 28: instead of “…increased as does the elevation.” Write “…increased with the elevation.”
- Review lines 28-32 for grammar and clarity.
Introduction
- Line 41: instead of “the composition change” write “the changes in composition”
- Lines 45-46: replace the expression “precipitation ion”, maybe just “precipitation”?
- Line 48: instead of “the leading factors” write “the leading pollutants”
- Line 49: instead of “and relatively backward” write “and is relatively backward”
- Line 50: instead of “and is less affected” write” and as such is less affected”
- Line 54: instead of “atmospher” write “atmosphere”
- Line 58: instead of “human settlement” write “human settlements”
- Lines 58-61: this sentence is very confusing, especially the part “As an important way of replenishing a river”, which is this way that you are referring to? How does this relate with the pH and EC which you refer afterwards?
Data and Method
- Line 106: instead of “low temperature” maybe use “freezing” here? Since temperature of -15 ºC is below freezing point for water.
- Line 108: instead of “the main ion concentration were detected” write “the main ion concentrations were determined”
- Line 110: instead of “EC and pH was determined” write “EC and pH were determined”
- Table 1: Why the difference in the number of samples in each place? Why not sample everyday (line 101- you refer you took 355 precipitation samples during one year) at all places? Is this due to the number of rainy days observed? Please provide more information that clarifies your sampling protocol.
- Line 126: instead of “the ith” write “the ith sample”
- Line 127: instead of “the ith precipitation” write “the ith sample”
- Lines 132-133: this sentence is not quite correct. The charge of an anion is not necessarily equal to that of a cation, but more that all negative anion charges must equal positive cation charges, for the charge conservation to hold. Please review the sentence adequately.
- I fail to see the point of the analysis provided in Fig. 2 when you admit in lines 137-139 that some ions were not taken into account. What was your goal when providing this information? It is a law of nature that electric charge must be conserved. Why confirm it, especially if not all the data is available?
- Is the neutralization factor method a novel formula? It cannot be found in the reference you provided.
- Line 188: instead of “ith” write “ith sample”
- Line 189: instead of “ith precipitation” write “ith sample”
Results
- Lines 203-205: This sentence is a bit confusing. Why is acid rain referred here? Especially since afterwards the range of pH you present is from 6.5 to 9, which is not in the acid range.
- Lines 210-211: The relationship between less precipitation, higher content of SO2, and lower pH is not clear. You could write a sentence that explains better this causality.
- Line 256: instead of “cation” write “cations”
- Lines 285-286: instead of “each successive year when precipitation increases gradually” write “each year precipitation increases gradually”
- Line 287: instead of “various ions decrease gradually” write “various ions decrease”
- Figure 6: what sampling point does this figure refer to? Or is this an average of all sampling points?
- Line 296-297: instead of “Regarding the ions” write “Regarding the other ions”
- Lines 299-301: this sentence is confusing, please review for better clarity.
- In Table 5, there is a clear difference in the trend from Dry season/Wet season between the Huajian sampling point and the other sampling points. I have not seen this addressed in the text, and I think it would be important to explain the reasons that lead to this difference.
- Lines 322-323: instead of “we can see the change in wet deposition in the research area under discussion, wet deposition is greatly affected”, write “we can see the change in wet deposition, in the research area under discussion, is greatly affected”.
- Line 331: instead of “is located near sampling site”, write “is located near the sampling site”.
- Line 333: instead of “leads to the large wet deposition” write “leads to large wet deposition”.
- Lines 348-351: this conclusion is not really reflected in the results, since EC seems actually constant except for Huajian sampling and not correlated with elevation. Also, the altitude laterality which you state in line 347 is represented in Fig. 1, it is not very clear in that figure, and we do not get the elevation values for the sampling points until later in the text (Table 7), so the statements you are giving now have no supporting data. The elevation information should be given earlier in the text, possibly close to Fig. 1.
- Besides the inconsistencies referred above, both graphs in Fig. 7 do not show a correlation between elevation and EC…and all that is written in the supporting text shows that elevated EC is probably derived from other factors that not elevation. So I question the pertinence of these graphs and the usefulness of the information here presented.
- The comparison analysis in lines 388-409 is oversimplified, especially in the beginning. E.g. in line 390 you state that “ion concentrations are higher”, however, this depends on the ion. There are many ions to be analysed and whose results are presented in Table 7 so a more detailed comparison should be given for this information to be properly interpreted and give viable conclusions.
- Line 436: instead of “is found distributes” write “is found distributed”
Conclusions
- Line 480: instead of “wet deposition N and S” write “wet deposition of N and S”
I hope you find my comments useful for the improvement of the manuscript.
